# PolyI:C Maternal Immune Activation on E9.5 Causes the Deregulation of Microglia and the Complement System in Mice, Leading to Decreased Synaptic Spine Density

**DOI:** 10.3390/ijms25105480

**Published:** 2024-05-17

**Authors:** Shuxin Yan, Le Wang, James Nicholas Samsom, Daniel Ujic, Fang Liu

**Affiliations:** 1Campbell Family Mental Health Research Institute, Centre for Addiction and Mental Health, 250 College St., Toronto, ON M5T 1R8, Canada; shuxin.yan@camh.ca (S.Y.); wangle@ojlab.ac.cn (L.W.); james.samsom@camh.ca (J.N.S.); daniel.ujic@mail.utoronto.ca (D.U.); 2Institute of Mental Health and Drug Discovery, Oujiang Laboratory (Zhejiang Lab for Regenerative Medicine, Vision, and Brain Health), School of Mental Health, Wenzhou Medical University, Ouhai District, Wenzhou 325000, China; 3Institutes of Medical Science, University of Toronto, 1 King’s College Cir., Toronto, ON M5S 1A8, Canada; 4Department of Psychiatry, University of Toronto, 250 College St., Toronto, ON M5T 1R8, Canada; 5Department of Physiology, University of Toronto, 1 King’s College Cir., Toronto, ON M5S 1A8, Canada

**Keywords:** maternal immune activation, synaptic pruning, dendritic spine density, microglia, complement

## Abstract

Maternal immune activation (MIA) is a risk factor for multiple neurodevelopmental disorders; however, animal models developed to explore MIA mechanisms are sensitive to experimental factors, which has led to complexity in previous reports of the MIA phenotype. We sought to characterize an MIA protocol throughout development to understand how prenatal immune insult alters the trajectory of important neurodevelopmental processes, including the microglial regulation of synaptic spines and complement signaling. We used polyinosinic:polycytidylic acid (polyI:C) to induce MIA on gestational day 9.5 in CD-1 mice, and measured their synaptic spine density, microglial synaptic pruning, and complement protein expression. We found reduced dendritic spine density in the somatosensory cortex starting at 3-weeks-of-age with requisite increases in microglial synaptic pruning and phagocytosis, suggesting spine density loss was caused by increased microglial synaptic pruning. Additionally, we showed dysregulation in complement protein expression persisting into adulthood. Our findings highlight disruptions in the prenatal environment leading to alterations in multiple dynamic processes through to postnatal development. This could potentially suggest developmental time points during which synaptic processes could be measured as risk factors or targeted with therapeutics for neurodevelopmental disorders.

## 1. Introduction

Since early epidemiological observations noticed that an increase in neurodevelopmental disorders coincided with seasonal outbreaks of a variety of infectious diseases, evidence has mounted linking infection during pregnancy with neuropsychiatric outcomes in offspring [1]. Recent meta-analyses show that infection or fever during early pregnancy increases the risk for multiple neurodevelopmental disorders, including autism spectrum disorder (ASD) [2,3], non-affective psychoses like schizophrenia [4], and attention deficit hyperactivity disorder (ADHD) [3], with additional studies also finding links to bipolar disorder, epilepsy, Parkinson’s disease, and cerebral palsy [5]. Non-infectious mechanisms of immune activation such as diabetes, rheumatoid arthritis, and other autoimmune disorders are also risk factors for ASD, schizophrenia, ADHD, and mood disorders [1]. Therefore, it is suggested that immune activation during an early critical period of fetal development can cause disruptions which permanently alter the developmental trajectory of the brain.

Maternal immune activation (MIA) animal models allow us to directly study how the maternal immune response affects development. A variety of immunogens are used to induce MIA, with a popular choice being the viral double-stranded RNA-mimicking molecule polyinosinic:polycytidylic acid (polyI:C) [6]. MIA offspring display disruptions in behaviors related to schizophrenia, anxiety, and ASD [7,8]. Variation in MIA protocols has led to complexity in the reported outcomes [7,9]. For example, early (mouse gestational day E9) but not late (E17) polyI:C exposure causes deficits in the schizophrenia-associated reflex behavior known as prepulse inhibition (PPI) and causes reductions in prefrontal dopamine D1 receptor expression, whereas late but not early exposure disrupted working memory and hippocampal NMDA-type glutamate receptors [10]. There are also exposure-timing differences in the neurodevelopmental effects of MIA, with late exposure (E17) leading to marked neuroanatomical changes and oxidative stress in E18 embryos but fewer detectable differences in adolescent and adult offspring, whereas early exposure (E9) led to increased E18 apoptotic cells with significant neuroanatomical differences in adolescent offspring [11,12]. Attention must be paid factors like the immunogen, exposure timing, dose, species and strain of animal when characterizing MIA models.

The brain’s resident immune cells, microglia, are key players in neurodevelopment, and may act as an important link between immune dysregulation in MIA and neurological outcomes [13]. Microglia populate the brain early in development (E9.5) and are replaced slowly, so early perturbations may carry forward [14,15,16]. In addition to supporting neuronal proliferation and survival [13], microglia play a key role in synaptic development through “pruning” synapses by mechanisms such as phagocytosis and trogocytosis [17,18,19]. Dysfunction in synaptic transmission is a key feature of schizophrenia and ASD, and MIA models recapitulate alterations in synaptic spine density; although, whether density is increased or decreased appears to be sensitive to brain region, sex, and the MIA protocol used [20].

Microglial synaptic pruning is regulated through the classical complement pathway, which is a cascade starting with C1q forming a complex to activate the C1s serine protease, which cleaves C4 and C2, ultimately leading to C3 activation, which signals for phagocytosis [21,22]. Mice deficient in complement components C3, C1q, and C4 have impaired synapse elimination in the retinogeniculate nucleus [18,23,24], and increased synaptic spine density [25,26]. Additionally, C4 overexpression causes spine dysgenesis [27]. A hypothesis for this process posits that C1q accumulates in synapses and forms a complex with postsynaptic density protein 95 (PSD95), which tags synapses for elimination [23,28,29,30]. Complex formation activates the classical pathway, leading to C3 cleavage, and signals for the phagocytosis of the tagged synapse via CR3 receptors on microglia [30]. Microglia and the complement system stand at the intersection between immune function and the nervous system, so are critical in understanding dysfunction in MIA models and the connection to neurodevelopment.

Previous reports on MIA provide snapshots of the consequences of MIA in offspring, typically measuring outcomes either early in postnatal development or in adulthood, when behavioral changes precipitate. However, to identify points in the developmental trajectory which ultimately lead to behavioral deficits, it is important to track how the dysregulation of important neurodevelopmental processes proceed over time. We used an MIA protocol, exposing pregnant CD-1 dams to polyI:C on E9.5, and measured outcomes throughout development at 2, 3, 6, 8, and 12 weeks of age with a focus on microglial synaptic pruning and complement signaling. E9.5 was chosen as previous reports using this exposure time showed greater neuroanatomical changes in offspring [10,11], and the timing is relevant for microglia development. Our data highlight long-term dysregulation, which occurs in early-exposure MIA, which could be relevant for identifying postnatal critical periods in synaptic development which may be targeted for bio-marker or therapeutic development.

## 2. Results

### 2.1. Maternal Immune Activation Deceased Synaptic Spine Density Starting from 3 Weeks of Age

Alterations in synaptic spine density are a key feature of multiple neurodevelopmental disorders which MIA attempts to model. We assessed basal dendritic spine density in layer II/III pyramidal neurons in the somatosensory cortices of saline- or polyI:C-exposed offspring (Figure 1A,B). There was a significant interaction effect between MIA exposure and age (F_4,765_ = 11.19, *p* < 0.00001, η^2^_p_ = 0.06 [0.03; 1]) on spine density (Figure 1C,D). Spine density was significantly lower in polyI:C offspring relative to saline controls starting from 3 weeks of age (wo) (2 wo, t(765) = 0.72, *p* = 0.47, d = 0.13 [−0.23; 0.49]; 3 wo, t(765) = 8.45, *p* < 0.00001, d = 1.18 [0.90; 1.46]; 6 wo, t(765) = 10.44, *p* < 0.00001, d = 1.64 [1.32; 1.96]; 8 wo, t(765) = 9.85, *p* < 0.00001, d = 1.58 [1.25; 1.90]; 12 wo, t(765) = 6.67, *p* < 0.00001, d = 1.17 [0.82; 1.51]). There were significant age effects in saline- (F_4,365_ = 6.00, *p* = 0.00011, η^2^ = 0.06 [0.02; 1]) and polyI:C-exposed offspring (F_4,400_ = 13.86, *p* < 0.00001, η^2^ = 0.12 [0.07; 1]). Spine density appeared to increase from 3 wo to 8 wo in saline controls (3 wo–6 wo, t(765) = −2.69, *p* = 0.021, d = −0.41 [−0.72; −0.11]; 3 wo–8 wo, t(765) = −4.013, *p* = 0.00026, d = −0.61 [−0.91; −0.31]) (see Figure 1D for all significant contrasts). However, spine density decreased in MIA offspring after 2 wo and remained reduced to 12 wo (2 wo–3 wo, t(765) = 7.14, *p* < 0.00001, d = 1.16 [0.84; 1.48]; 2 wo–12 wo t(765) = 4.98, *p* < 0.00001, d = 0.88 [0.53; 1.23]). This suggests that MIA caused persistent reductions in dendritic spine density which begin at 3 weeks of age and continue into adulthood.

Controlling for animal ID, the MIA exposure × age interaction just misses the significance threshold (F_4,19_ = 2.86, *p* = 0.053, η^2^_p_ = 0.38 [0; 1]), but a main effect of MIA remains significant (F_1,19_ = 49.91, *p* < 0.00001, η^2^_p_ = 0.73 [0.52; 1]). MIA-related decreases in spine density past 2 wo remained significant at all ages (3 wo p_adj_ = 0.0077, 6 wo p_adj_ = 0.00038, 8 wo p_adj_ = 0.00041, 12 wo p_adj_ = 0.012). Age effects in saline controls lose significance (F_4,9_ = 1.37, *p* = 0.31, η^2^ = 0.31 [0; 1]) but remain significant in polyI:C offspring (F_4,9_ = 3.92, *p* = 0.31, η^2^ = 0.64 [0.03; 1]). This suggests that the effects of MIA treatment in our sample were not biased by the individuals sampled for each dendrite, but age effects were not robust in the saline group.

### 2.2. Maternal Immune Activation Increased Microglial Synaptic Pruning in an Age-Dependent Manner

Microglia play a key role in regulating synaptic spines by pruning spines marked for elimination [17,18,19]. We performed a co-staining of the postsynaptic marker PSD95 (protein postsynaptic density protein 95) and the microglial protein Iba1 (Ionized calcium-binding adaptor molecule 1) at different time points throughout development in the mouse somatosensory cortex. PSD95 overlapping with Iba1 indicates synaptic spines that are in contact or within microglia, which is a correlate for microglial synaptic pruning. There was a significant interaction between MIA and age on the fraction of PSD95 and Iba1 co-staining (F_4,221_ = 5.46, *p* = 0.00032, η^2^_p_ = 0.09 [0.03; 1]) (Figure 2). Post hoc contrasts showed significantly increased co-staining at 3 wo (t(221) = 2.72, *p* = 0.021, d = 0.79 [0.21; 1.36]), 6 wo (t(221) = 6.53, *p* < 0.00001, d = 1.74 [1.19; 2.29]) and 8 wo (t(221) = 3.98, *p* = 0.00024, d = 1.34 [0.67; 2.02]) in polyI:C relative to saline offspring. Age effects were significant in both saline (F_4,108_ = 4.31, *p* = 0.0028, η^2^ = 0.14 [0.03; 1]) and polyI:C groups (F_4,113_ = 16.70, *p* < 0.00001, η^2^ = 0.37 [0.25; 1]). Saline-exposed offspring had significantly decreased co-staining at 8 wo (3 wo–8 wo, t(221) = 3.31, *p* = 0.0027, d = 1.016 [0.43; 1.61]) and 12 wo (3 wo–8 wo, t(221) = 3.44, *p* = 0.0022, d = 1.034 [0.41; 1.66]) relative to the youngest age groups, with the highest point at 3 wo, whereas polyI:C offspring had a clear peak shifted to 6 wo (2 wo–6 wo, t(221) = −3.86, *p* = 0.00099, d = −1.069 [−1.62; −0.51]), with decreases from 8 wo (6 wo–8 wo, t(221) = 3.41, *p* = 0.0022, d = 1.023 [0.42; 1.62]) to 12 wo (6 wo–12 wo, t(221) = 5.54, *p* < 0.00001, d = 2.16 [1.59; 2.74]) (see Figure 2B for all significant contrasts). These data indicate that polyI:C-induced MIA causes increased microglia-mediated synaptic pruning, and possibly shifts the developmental pattern of peak pruning to a later age.

When controlling for animal ID, the main effects of MIA treatment (F_4,12_ = 30.14, *p* = 0.00012, η^2^_p_ = 0.71 [0.42; 1]) and age (F_4,12_ = 10.69, *p* = 0.00056, η^2^_p_ = 0.78 [0.48; 1]) retain significance; however, the interaction just misses significance (F_4,12_ = 3.13, *p* = 0.054, η^2^_p_ = 0.50 [0; 1]). MIA-related increases in Iba1-PSD95 co-staining lost significance at 3 wo (p_adj_ = 0.14) but remained significant at 6 wo (p_adj_ = 0.0039) and 8 wo (p_adj_ = 0.024). Differences across age remained significant in in polyI:C offspring (F_4,8_ = 7.50, *p* = 0.012, η^2^ = 0.83 [0.34; 1]), but not in saline controls (F_4,9_ = 3.22, *p* = 0.11, η^2^ = 0.70 [0; 1]). This suggests MIA-related increases in age-dependent microglial synaptic pruning were not biased by the animals sampled, though overall age differences within the saline group were not as robust.

### 2.3. Maternal Immune Activation Increased CD68 Protein Expression in Microglia in an Age-Dependent Manner

While PSD95-Iba1 overlap is evidence for microglia–synapse interaction, it does not directly suggest a mechanism for synapse regulation. CD68 (cluster of differentiation 68) is a marker of phagocytic microglia [33]. To measure phagocytosis, we quantified CD68 staining within Iba1^+^ cells at different developmental points in the somatosensory cortices of polyI:C-exposed and control offspring. CD68-Iba1 co-staining was significantly affected by MIA treatment × age interactions (F_4,226_ = 13.88, *p* < 0.00001, η^2^_p_ = 0.20 [0.12; 1]) (Figure 3). Post hoc contrasts showed MIA offspring had significantly elevated Iba1-CD68 co-staining at all ages except 2 wo (2 wo, t(226) < 0.001, *p* = 0.99, d < 0.001 [−60; 0.60]; 3 wo, t(226) = 4.20, *p* < 0.00001, d = 1.24 [0.65; 1.83]; 6 wo, t(226) = 11.13, *p* < 0.00001, d = 2.90 [2.32; 3.48]; 8 wo t(226) = 6.44, *p* < 0.00001, d = 1.92 [1.31; 2.53]; 12 wo, t(226) = 4.39, *p* < 0.00001, d = 1.36 [0.74; 1.99]). Age effects were significant in both control (F_4,112_ = 65.20, *p* < 0.00001, η^2^ = 0.70 [0.62; 1]) and MIA offspring (F_4,114_ = 7.34, *p* < 0.00001, η^2^ = 0.20 [0.09; 1]). In saline controls, there was an immediate reduction from 2 wo to 3 wo (2 wo–3 wo, t(226) = 9.11, *p* < 0.00001, d = 2.69 [2.05; 3.32]), and levels remained lower until 12 wo (2 wo–12 wo, t(226) = 10.033, *p* < 0.00001, d = 2.99 [2.34; 3.64]), though the lowest levels were at 6 wo (3 wo–6 wo, t(226) = 2.57, *p* = 0.022, d = 0.74 [0.17; 1.30]) (see Figure 3B for all significant contrasts). In polyI:C-exposed offspring, levels drop from the high point at 2 wo (2 wo–3 wo, t(226) = 4.74, *p* < 0.00001, d = 1.45 [0.83; 2.07]) but rise again at 6 wo (3 wo–6 wo, t(226) = −3.43, *p* = 0.0018, d = −0.93 [−1.46; −0.39]) before tapering off (6 wo–12 wo, t(226) = 3.95, *p* = 0.00030, d = 1.11 [0.55; 1.68]). This suggests that prenatal polyI:C exposure dysregulated age-related reductions in phagocytic activity within microglia, particularly at 6 wo, leading to elevated activity past 2 wo relative to control mice.

The MIA treatment × age interaction remained significant when controlling for animal ID (F_4,12_ = 6.72, *p* = 0.0048, η^2^_p_ = 0.70 [0.29; 1]). MIA-related increases in Iba1-CD68 co-staining remained significant at all ages past 2 wo (3 wo p_adj_ = 0.019, 6 wo p_adj_ < 0.00001, 8 wo p_adj_ = 0.0021, 12 wo p_adj_ = 0.011). Age effects remained significant in control (F_4,7_ = 41.86, *p* = 0.00048, η^2^ = 0.97 [0.87; 1]) but not MIA offspring (F_4,7_ = 3.78, *p* = 0.059, η^2^ = 0.68 [0; 1]). This confirms that MIA effects on phagocytic activity in microglia were not biased by animal sampling, and reinforces that MIA may prevent the reduction in phagocytic microglia after 2 wo seen in control animals.

### 2.4. Maternal Immune Activation Affects PSD95 and Complement Protein Expression in an Age-Dependent Manner

The complement system plays an important role in synaptic spine regulation [21]. We tracked the protein expression of important complement components, their activated cleaved forms, and PSD95 throughout development in MIA and control offspring. Blots were normalized to the control means to compare MIA exposure at each age group. PSD95 expression was significantly lower in 3 wo MIA offspring (t(5.7) = 3.89, *p* = 0.045, d = 2.75 [1.79, 8.31]) (Figure 4B). Lower PSD95 was seen in 2 wo MIA offspring with a large effect, but significance did not survive multi-test correction (t(5.7) = 2.80, *p* = 0.033, p_adj_ = 0.13, d = 1.86 [0.58, 8.28]). Complement C1q expression was significantly higher in polyI:C-exposed offspring at 8 wo (t(7.5) = −3.45, *p* = 0.049, d = −2.18 [−4.65, −1.64]) (Figure 4C). Complement C4 levels were also significantly elevated in 8 wo MIA mice (t(7.2) = −5.48, *p* = 0.0043, d = −3.47 [−9.70, −2.22]) (Figure 4D). The C4 α-chain did not significantly differ between groups; however, increased expression in 12 wo MIA offspring was significant before multi-test correction (t(6.0) = −2.94, *p* = 0.026, p_adj_ = 0.13, d = −2.08 [−17.20, −0.85]) (Figure 4E). Complement C4d expression did not differ with MIA (Figure 4F). Due to a small variance, differences in the 3 wo group approached significance (t(4.5) = −4.10, *p* = 0.012, p_adj_ = 0.059, d = −2.90 [−11.60, −1.68]). There also appeared to be elevated C4d levels at 8 wo, mirroring C1q and C4; however, these differences were not significant (t(7.3) = −1.70, *p* = 0.13, p_adj_ = 0.53, d = −1.07 [−2.64, −0.03]). Full-length complement component C3, C3 α-chain, and C3 α-chain fragment expression were not significantly affected by MIA at any age (Figure 5). Overall, there was evidence of an age-dependent dysregulation of PSD95, complement C1q, and complement C4 protein expression.

## 3. Discussion

We measured microglial synaptic spine pruning and complement signaling at different ages in order to better understand the altered developmental trajectory in the E9.5 polyI:C MIA mouse model for neurodevelopmental disorders. PolyI:C-exposed offspring had reductions in dendritic spine density within the cerebral cortex starting from 2 weeks of age, and there were requisite increases in microglial synaptic pruning and phagocytic microglia. Additionally, we have shown evidence of dysregulated complement C4 signaling throughout development.

The impact of MIA on synaptic spine density is complex. In line with our findings, decreased spine density was shown in the somatosensory cortex of adolescent and adult mice using polyI:C MIA [34]. Decreased spine density was also shown in the hippocampus of mhDISC1 transgenic mice [35], as well as in adult, but not juvenile, LPS MIA rats [36,37]. Contrary to our results, increased spine density was seen in 8–9-week-old polyI:C-exposed mouse prefrontal and somatosensory cortex [38,39], and in hippocampal granule cells of 15-day-old LPS-exposed mice [40]. Strain differences may exist, as groups finding increases used C57BL/6 mice [38,39] and those finding decreases used genetically modified mice (Thy1-YFP-H and mhDISC1) [34,35], while we used CD-1 mice. One group found spine density was increased or decreased depending on sex and whether the synapses were excitatory or inhibitory [41]. We did not test female mice, nor could we differentiate synapse type, so these factors may play a role. Interestingly, spine density is decreased in schizophrenia and increased in ASD in human post mortem samples [20]. It is possible that there is some early hidden variable which tips the dysregulation of spine dynamics in one direction or the other, leading to either a schizophrenia-like or autism-like phenotype in MIA models.

Our results suggest increased overall synapse engulfment by microglia, as well as increased microglial phagocytosis; therefore, the mechanism leading to reduced spine density may be a dysregulation of microglia, leading to increased pruning. Interestingly, despite showing increased spine density, a previous study found increased microglia-spine interactions using 3D reconstruction analysis at a voxel resolution of 0.2 × 0.2 × 0.3 µm to measure neuronal signals engulfed within Iba1-stained microglia, which supports our findings [38]. We have performed this same type of PSD95-Iba1 co-localization analysis using super-resolution microscopy at 0.2 µm z-steps and found no difference between the 3D analysis and the more conventional 2D analysis used in this study, so we argue that increased resolution would be unlikely to affect our results [30]. Other groups found the downregulation of some phagocytosis-associated genes [40,42]; although, the direct link between the expression of these genes and synaptic pruning is less established. Consistent with our results, increased CD68 in microglia was shown previously [41]. Notably, we found that MIA disrupted age-specific patterns in microglial pruning and phagocytosis. It is possible that this shift in the normal timing of microglial synaptic regulation could influence how a prenatal environmental exposure can underlie conditions, like schizophrenia, which have onsets in early adulthood. However, pruning and phagocytosis did not perfectly track each other in our data; though, this is not necessarily surprising, as microglia have multiple functions aside from synapse regulation. Spine density also did not perfectly track the timing of microglial synaptic pruning. Microglial pruning is balanced by spine morphogenesis, so this equilibrium may be what is shifted by MIA. This could be tested by imaging spine dynamics in live mice, taking advantage of the Thy1-YFP-H mouse line to visualize spine dynamics as they occur in real-time.

Our data suggest a dysregulation of the complement system. Treatment differences in C4 and its activated cleaved products were seen at 8 weeks of age, after the peak in microglial synaptic pruning. Compensatory mechanisms for early low levels of C4 could cause an overshoot in complement-regulated synaptic pruning at later ages, which supports a mechanism whereby early dysregulation can precipitate late-onset changes. Still, we did not see a linear correspondence between complement, microglial pruning, and spine density. By measuring the protein levels in the whole brain, we may have washed out region-specific signals. Previous groups found differences in spine dynamics relative to region and neuron type [20]. Complement expression changes in affected synapses isolated to specific regions may be proportionately masked when including large amounts of brain tissue from regions where no changes are occurring. This may also explain a lack of significant differences seen in C3. C3 activation is the convergence point of the classical, lechtin, and alternative complement pathways [21]. As C3 is participating in many functions other than synaptic pruning, alterations in C3 especially may not be detectable at the level of the entire brain, and may only be appear in certain regions or even synapse types. Analyzing specific brain regions, or using single-cell sequencing, could be informative. There are few reports of complement expression in MIA; however, previous groups have found either increased C1q or decreased C1q and C3 in adult MIA offspring [41,43]. Interestingly, there is evidence of an overactive classical pathway in schizophrenia, including the overexpression of C1q, C3, and C4 [44]; conversely, there is weak evidence of reduced levels of these genes in ASD [45]. Hence, complement signaling may be another balancing point where dysregulation can tip in either direction, leading to two different pathologies; although, the data are considerably noisier when relating the complement system to either schizophrenia or ASD.

MIA is an important model for environmental factors influencing neuropsychiatric disorders, but is sensitive to subtle methodological differences. As we found decreased spine density, and increased C4 and C4 activation in E9.5 polyI:C-exposed CD-1 offspring, this specific protocol is more suited as a model for schizophrenia than ASD. Future research should focus on untangling the mechanisms by which MIA apparently leads to divergent outcomes such as increased vs. decreased spine density. We found many disruptions peaked in late adolescence and early adulthood, which could suggest that postnatal interventions, even late in life, may have corrective effects in MIA. Treatments which regulate microglia or inflammation starting after 3 weeks of age should be explored in rescuing MIA deficits. Furthermore, our data hints that the bio-markers of neuroinflammation or synaptic regulation measured in late adolescence could be predictive for schizophrenia risk. MIA as a model will continue to produce important insights into the links between immune function and neurodevelopment.

## 4. Materials and Methods

### 4.1. Animals and Maternal Immune Activation Model

Seven-week-old CD-1 mice were purchased from Charles River Laboratories and acclimatized to the housing facility for 1 week. Mice pairs bred and pregnancy was assessed by plug checking, with plug discovery denoting gestational day (E) 0.5. On E9.5, dams received an acute intraperitoneal (i.p.) injection of saline or polyinosinic:polycytidylic acid (polyI:C) (20 mg/kg, Sigma, catalog P1530). Litters were weaned on postnatal day (P) 21 and groups were housed with their littermates (2–5 mice per cage, 20–23 °C, 12 h light/dark cycle). In total, 2–3 new breeding pairs per treatment were used to generate independent age cohorts assessed at 2, 3, 6, 8 and 12 weeks of age. The same mouse was not assessed at multiple ages. Male offspring were used for the analysis. All animal procedures were approved by the Animal Care Committee at the Centre for Addiction and Mental Health (CAMH, Toronto, CA, USA) in accordance with the Canadian Commission on Animal Care (CCAC).

### 4.2. Immunohistochemistry

Animals were sacrificed by anesthetic overdose (isoflurane) and transcardially perfused with phosphate-buffered saline (PBS) followed by 4% paraformaldehyde (PFA, in PBS). Brains were post-fixed in 4% PFA overnight, then cryoprotected in 30% sucrose until saturated before freezing at −80 °C. Frozen coronal sections (25 µm) were cut using a cryostat. Sections were permeabilized with 0.3% Triton X-100 in PBS for 30 min and blocked with 2% bovine serum albumin in PBS at room temperature for 1.5 h. Sections were incubated with anti–Iba1 antibody (1:200, Novus, catalog NB100-1028) and PSD95 (1:300, SYSY, catalog 124008) or CD68 (1:200, Bio-Rad, catalog MCA1957) overnight at 4 °C. Secondary antibodies were probed for 1.5 h at room temperature (Invitrogen, Waltham, MA, USA, Alexa Fluor^TM^ Plus 594 donkey anti-goat IgG (H+L), catalog #A32758; Alexa Fluor^TM^ Plus 488 donkey anti-rabbit IgG (H+L), catalog #A32790; Alexa Fluor^TM^ Plus 488 donkey anti-rat IgG (H+L), catalog #A48269). All Images were acquired using an Olympus FluoView FV1200 confocal microscope at 63 × magnification using 1.0µm z-steps. Images were randomly sampled from the somatosensory cortex using the rough brain morphology as landmarks to choose appropriate sections (Figure 1A). Signals outside of the microglia were filtered from the image using Threshold Color in ImageJ (Version ij153-win-java8, National Institutes of Health), then analyzed by M1 and M2 coefficients using the JACoP plugin to calculate the summed intensities of pixels from the green image for which the intensity in the red channel is above zero.

### 4.3. Golgi Staining

Mice were sacrificed by cervical dislocation and whole brains were dissected for Golgi staining according to the manufacturer’s instructions (FD Rapid GolgiStain™ Kit, FD Neurotechnologies, catalog PK401). Coronal sections (200 μm) were prepared using a vibratome. Images were acquired using an Olympus FluoView FV1200 confocal microscope, brightfield illumination at 100× magnification. Basal dendrites were imaged from randomly-chosen layer II/III pyramidal neurons located in the somatosensory cortex identified using rough brain morphology (Figure 1A,B). One dendrite segment was analyzed per imaged neuron (mean length 45.6 µm ± 14.7 µm). Dendrite segments were measured in ImageJ, then spines were counted manually.

### 4.4. Protein Extraction and Western Blots

Mice were sacrificed by cervical dislocation, then whole brains were removed and flash frozen on dry ice before being stored at −80 °C. Brain tissue was lysed in RIPA buffer (50 mM Tris, 150 mM NaCl, 2 mM EDTA, 0.5% sodium deoxycholate, 1% NP40, 1% Triton X-100, 0.1% SDS, and pH 7.4) with protease inhibitors at 4 °C for 1 h. Cell debris was pelleted by centrifugation at 14,000× *g* for 15 min at 4 °C, and the supernatants were collected. Tissue extracts were denatured in Laemmli buffer for 5 min at 95 °C, subjected to SDS-PAGE, and transferred onto nitrocellulose. Membranes were blocked for 1 h at room temperature with 5% nonfat dry milk (*w*/*v*) in TBST (0.1% Tween 20), and incubated with primary antibodies overnight at 4 °C, followed by secondary antibodies for 1 h at room temperature with 4 washes after each incubation in TBST. Blots were scanned with a ChemiDoc™ MP Imaging System (Bio-Rad Laboratories, Hercules, CA, USA) and quantified with Image Lab software (Version 5.2.1, Bio-Rad). The primary antibodies used were rabbit-monoclonal anti-PSD95 (Abcam, Waltham, MA, USA, catalog ab238513), rabbit-polyclonal anti-C1q (Abcam, catalog ab155052), rabbit-monoclonal anti-C3 (Abcam, catalog ab200999), rat-monoclonal anti-C4b/d (Novus Biologicals, Centennial, CO, USA, catalog NB200-541), and mouse-monoclonal anti-β-actin (Sigma, Oakville, ON, Canada, catalog A5441). The secondary antibodies used were HRP-linked anti-rabbit IgG (CST, catalog 7074S), HRP-linked anti-mouse IgG (CST, catalog 7076S), and HRP-linked anti-rat IgG (CST, catalog 7077S).

### 4.5. Statistical Analysis

Parametric assumptions were checked with Shapiro–Wilk’s test for normality, Bartlett’s test for equal variance, 3× the interquartile range for extreme outliers, and by examining residual plots. Data meeting parametric assumptions were analyzed using 2-way ANOVAs. Post hoc pairwise contrasts were compared with either false-discovery rate or Tukey’s method for multi-test correction. Post hoc 1-way ANOVAs were used for overall age effects. Animal ID was included as a random effect for re-analysis for hierarchical data, where multiple samples were taken from the same animal. Western data was compared with Welch’s *t*-tests using Holm’s method for multi-test correction. Confidence intervals for effect sizes where calculated using bootstrap estimation. All analyses were conducted in R using the ‘base’, ‘effectsize’, ‘rstatix’, ‘emmeans’, ‘lme4’, and ‘lmertest’ packages.

## Figures and Tables

**Figure 1 ijms-25-05480-f001:**
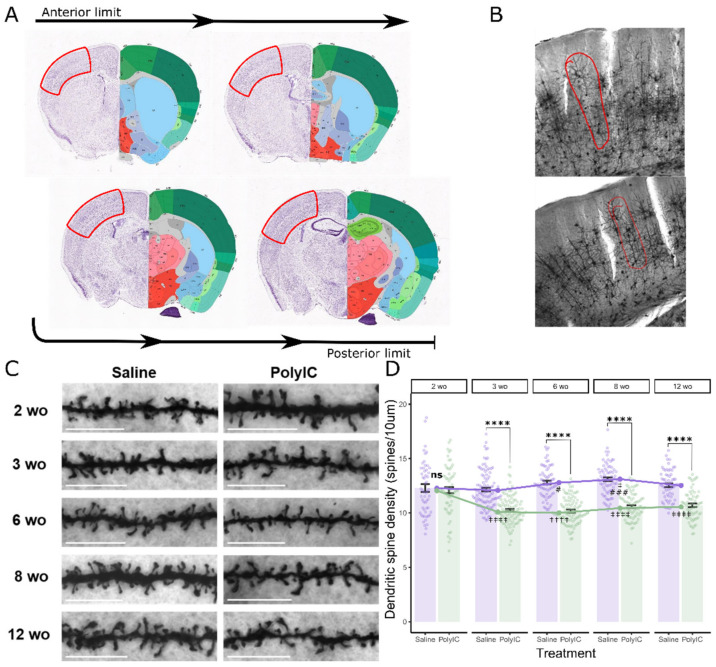
Maternal immune activation deceased spine density from 3 weeks of age to adulthood. (**A**) Representative slices showing the brain regions sampled for imaging analysis. Idealized morphology of the anterior-most and posterior-most coronal slices sampled is shown. The cortex region of interest (ROI) is circled in red. Images downloaded from the Allen Brain Atlas (https://mouse.brain-map.org/experiment/thumbnails/100048576?image_type=atlas (accessed on 21 November 2023)) [31,32]. (**B**) Representative Golgi images showing layer II/III cortical pyramidal neurons. Neurons were randomly sampled within the red circled area. (**C**) Representative images of basal dendrites from Golgi-stained layer II/III pyramidal neurons sampled from the somatosensory cortex in offspring of saline- or polyI:C-treated dams at 2, 3, 6, 8, and 12 weeks of age (wo). Scale bar, 10 µm. (**D**) Quantification of cortical dendritic spine density (mean analyzed dendritic segment length was 45.6 µm ± 14.7 µm (SD); number of dendrites: saline: 2 wo n = 57, 3 wo n = 104, 6 wo n = 70, 8 wo n = 75, 12 wo n = 64; PolyIC: 2 wo n = 61, 3 wo n = 101, 6 wo n = 95, 8 wo n = 81, 12 wo n = 67; from n = three–four mice in each group). Error bars represent the mean ± SEM, two-way aligned-ranks ANOVA, fdr-corrected post hoc marginal means, comparison indicators (ns, ns-not significant, treatment effect comparison relative to saline group, ‡ age effect comparison relative to 2 wo, # relative to 3 wo), ns-not significant, ^‡^, ^#^ *p* < 0.05, ^###^ *p* < 0.001, ^****^, ^††††^, ^‡‡‡‡^ *p* < 0.0001. Raw data tables are contained in Appendix A.

**Figure 2 ijms-25-05480-f002:**
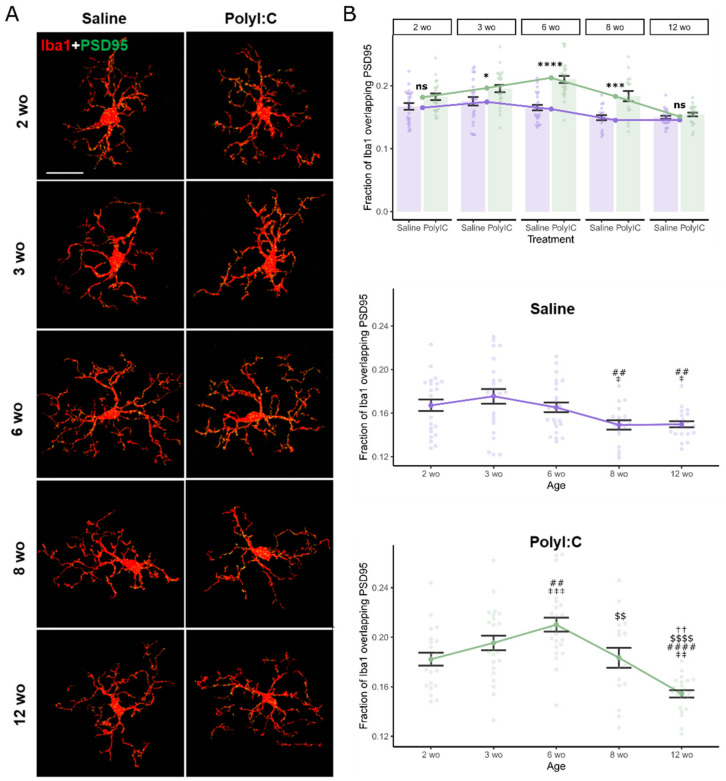
Maternal immune activation increased microglia–synapse overlap in an age-dependent manner. (**A**) Representative confocal images of antibody staining against microglia marker Iba1 (red) and synapse marker PSD95 (green) in offspring of saline- or polyI:C-treated dams at 2, 3, 6, 8, and 12 weeks of age (wo). Scale bar, 20 µm. (**B**) Quantification of the fraction of Iba1 overlapping PSD95 (number of views analyzed: saline: 2 wo n = 24, 3 wo n = 24, 6 wo n = 25, 8 wo n = 18, 12 wo n = 22; PolyIC: 2 wo n = 22, 3 wo n = 24, 6 wo n = 32, 8 wo n = 17, 12 wo n = 23; from n = 2–3 mice in each group). Treatment-effect comparison indicated as * relative to saline group, separate trend-lines for each treatment across age are shown below, ‡ relative to 2 wo, # relative to 3 wo, $ relative to 6 wo, † relative to 8 wo. Error bars represent the mean ± SEM, two-way aligned-ranks ANOVA, fdr-corrected post hoc marginal means, ns-not significant, *, ^‡^ *p* < 0.05, ^‡‡^, ^##^, ^$$^, ^††^ *p* < 0.01, ***, ^‡‡‡^ *p* < 0.001, ****, ^####^, ^$$$$^ *p* < 0.0001. Raw data tables are contained in Appendix A.

**Figure 3 ijms-25-05480-f003:**
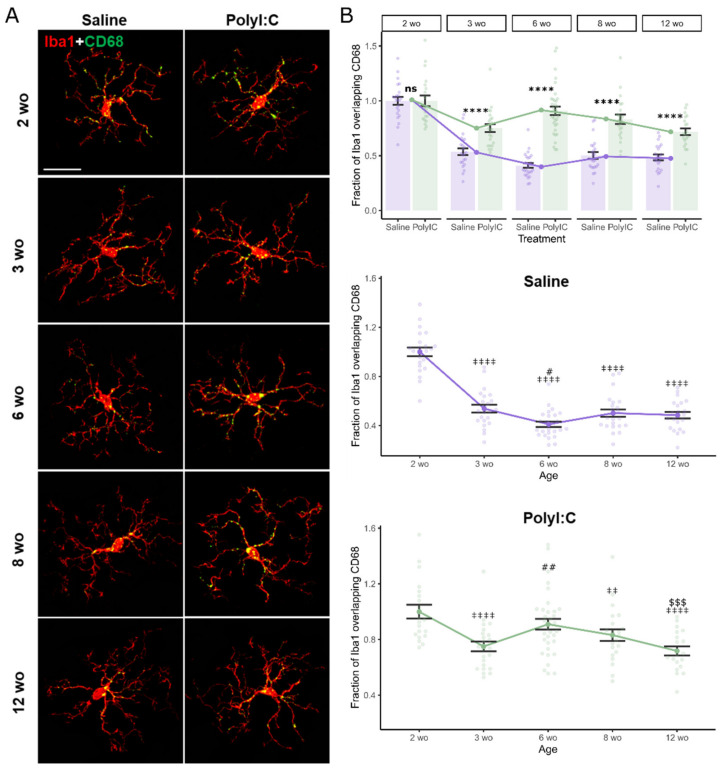
Maternal immune activation increased expression of the microglia phagocytosis marker CD68 in an age-dependent manner. (**A**) Representative confocal images of antibody staining against microglia marker Iba1 (red) and phagocytosis marker CD68 (green) in offspring of saline- or polyI:C-treated dams at 2, 3, 6, 8, and 12 weeks of age (wo). Scale bar, 20 µm. (**B**) Quantification of the fraction of Iba1 overlapping CD68 (number of views analyzed: saline: 2 wo n = 23, 3 wo n = 23, 6 wo n = 25, 8 wo n = 23, 12 wo n = 22; PolyIC: 2 wo n = 20, 3 wo n = 23, 6 wo n = 33, 8 wo n = 22, 12 wo n = 20; from n = 2–3 mice in each group). Treatment-effect comparison indicated as * relative to saline group, and separate trend-lines for each treatment across age are shown below, ‡ relative to 2 wo, # relative to 3 wo, $ relative to 6 wo. Error bars represent the mean ± SEM, two-way ANOVA, fdr-corrected post hoc marginal means, ns-not significant, ^#^ *p* < 0.05, ^‡‡^, ^##^ *p* < 0.01, ^$$$^ *p* < 0.001, ****, ^‡‡‡‡^ *p* < 0.0001. Raw data tables are contained in Appendix A.

**Figure 4 ijms-25-05480-f004:**
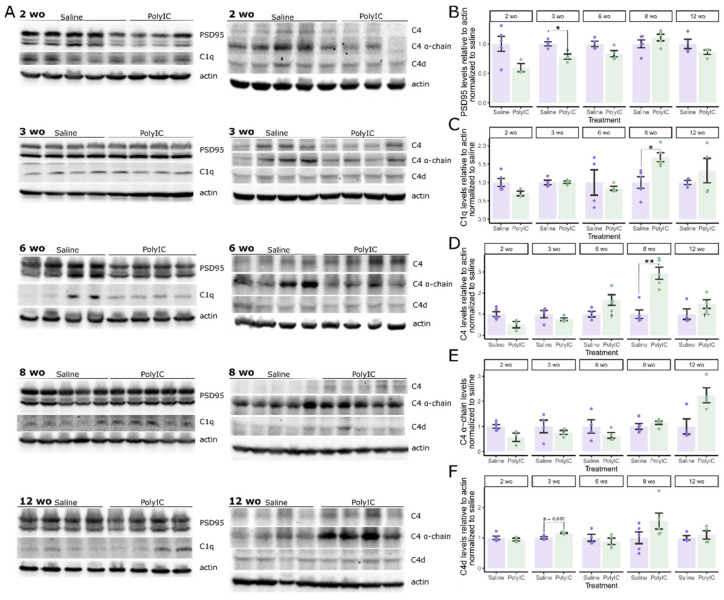
Maternal immune activation affects PSD95, and complement C1q and C4 protein expression in an age-dependent manner. Protein levels were measured in offspring of saline- or polyI:C-treated dams at 2, 3, 6, 8, and 12 weeks of age (wo). (**A**) Representative Western blots (saline: 2 wo n = 5, 3 wo n = 4, 6 wo n = 4, 8 wo n = 5, 12 wo n = 4; PolyIC: 2 wo n = 3, 3 wo n = 4, 6 wo n = 4, 8 wo n = 5, 12 wo n = 4). (**B**) Densitometry analysis of PSD95 protein levels. (**C**) Densitometry analysis of complement C1q protein levels. (**D**) Densitometry analysis of complement C4 levels. (**E**) Densitometry analysis of C4 α-chain levels. (**F**) Densitometry analysis of complement C4d levels. Protein levels are expressed relative to actin and normalized to the mean levels in the saline control group for each age. Error bars represent the mean ± SEM, * *p* < 0.05, ** *p* < 0.01, Holm-corrected Welch’s *t*-tests. Raw data tables, raw Western blot images, and annotated summary file of Western blot images are contained in Appendix A.

**Figure 5 ijms-25-05480-f005:**
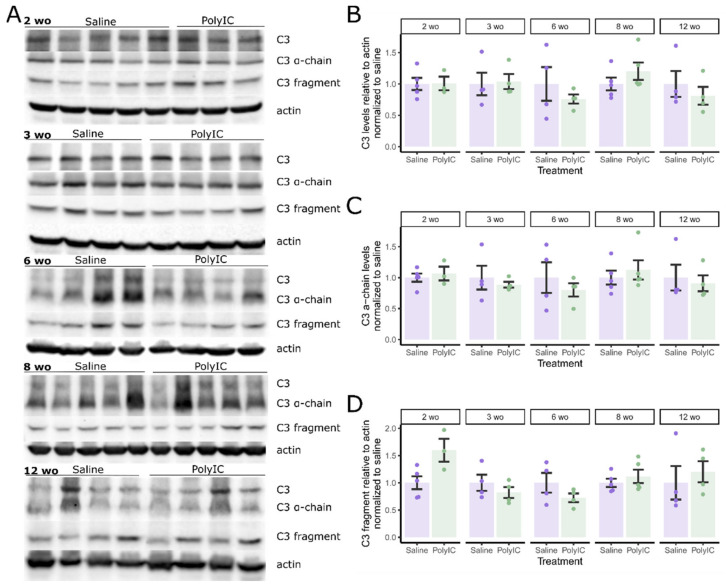
Maternal immune activation has no effect on complement C3 protein expression. C3 and activated fragment levels were measured in offspring of saline- or polyI:C-treated dams at 2, 3, 6, 8, and 12 weeks of age (wo). (**A**) Representative Western blots (saline: 2 wo n = 5, 3 wo n = 4, 6 wo n = 4, 8 wo n = 5, 12 wo n = 4; PolyIC: 2 wo n = 3, 3 wo n = 4, 6 wo n = 4, 8 wo n = 5, 12 wo n = 4). (**B**) Densitometry analysis of full length C3 protein levels. (**C**) Densitometry analysis of C3 α-chain protein levels. (**D**) Densitometry analysis of C3 fragment levels. Protein levels are expressed relative to actin and normalized to the mean levels in the saline control group for each age. Error bars represent the mean ± SEM, Holm-corrected Welch’s *t*-tests. Raw data tables, raw Western blot images, and annotated summary file of Western blot images are contained in Appendix A.

## Data Availability

The original contributions presented in the study are included in the article/Appendix A, and further inquiries can be directed to the corresponding authors.

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
