# Peer review of "PolyI:C Maternal Immune Activation on E9.5 Causes the Deregulation of Microglia and the Complement System in Mice, Leading to Decreased Synaptic Spine Density"

_ijms, 2024, doi:10.3390/ijms25105480_

Round 1

Reviewer 1 Report

Comments and Suggestions for Authors

The authors presented the results of a study indicating a significant effect of immune activation during pregnancy on dendritic spike density and phagocytic activity of microglia further linked to changes in the expression of some complement components at different times in the offspring.

Minor comments

I propose to cite more information on the involvement of the complement system in the „pruning” process. The authors only mention the classical pathway of the complement system in the synapse deletion process discussed.

How do the authors interpret the changes in C1q and C4 levels and the lack of change in C3 protein expression in the context of their contribution to the phagocytic activity of microglia in „pruning” process?

In my opinion, the components from the statistical analysis placed in the supplementary materials will improve the reading of the description of the results.

Description to Figure 3. I would suggest changing the title to: MIA activation increased expression of the microglia phagocytosis marker CD68 in an age-dependent manner.

Author Response

We thank the reviewer for their helpful comments and suggestions. Our response and edits are highlighted below:

I propose to cite more information on the involvement of the complement system in the „pruning” process. The authors only mention the classical pathway of the complement system in the synapse deletion process discussed.

We have expanded the background on the involvement of the classical complement pathway in synaptic pruning in the introduction (lines 75-86). It now reads:

“Microglial synaptic pruning is regulated through the classical Complement pathway, which is a cascade starting with C1q forming a complex to activate the C1s serine protease which cleaves C4 and C2 ultimately leading to C3 activation which signals for phagocytosis [21, 22]. Mice deficient in complement components C3, C1q, and C4 have impaired synapse elimination in the retinogeniculate nucleus [18, 23, 24], and increased synaptic spine density [25, 26]. Additionally, C4 overexpression causes spine dysgenesis [27]. A hypothesis for this process posits that C1q accumulates in synapses and forms a complex with postsynaptic density protein 95 (PSD95) which tags synapses for elimination [23, 28-30]. Complex formation activates the classical pathway leading to C3 cleavage and signals for phagocytosis of the tagged synapse via CR3 receptors on microglia [30]. Microglia and the complement system stand at the intersection between immune function and the nervous system, so are critical in understanding dysfunction in MIA models and the connection to neurodevelopment.”

Additional references include:

  1. Stephan, A. H.; Barres, B. A.; Stevens, B., The complement system: an unexpected role in synaptic pruning during development and disease. Annu Rev Neurosci 2012, 35, 369-89.
  2. Datta, D.; Leslie, S. N.; Morozov, Y. M.; Duque, A.; Rakic, P.; van Dyck, C. H.; Nairn, A. C.; Arnsten, A. F. T., Classical complement cascade initiating C1q protein within neurons in the aged rhesus macaque dorsolateral prefrontal cortex. J Neuroinflammation 2020, 17, (1), 8-8.
  3. Dejanovic, B.; Huntley, M. A.; De Mazière, A.; Meilandt, W. J.; Wu, T.; Srinivasan, K.; Jiang, Z.; Gandham, V.; Friedman, B. A.; Ngu, H.; Foreman, O.; Carano, R. A. D.; Chih, B.; Klumperman, J.; Bakalarski, C.; Hanson, J. E.; Sheng, M., Changes in the Synaptic Proteome in Tauopathy and Rescue of Tau-Induced Synapse Loss by C1q Antibodies. Neuron 2018, 100, (6), 1322-1336.e7.

How do the authors interpret the changes in C1q and C4 levels and the lack of change in C3 protein expression in the context of their contribution to the phagocytic activity of microglia in „pruning” process?

This is a good question. We measured complement expression in protein extracts from the entire mouse brain. C3 cleavage is a common step shared in all 3 complement pathways (classical, lechtin, and alternative). As C3 is involved in many processes in addition to synaptic pruning, it may be the case that any signal from C3 or C3 cleavage protein expression was washed out by sampling from the entire brain. We have added this explanation to the discussion (lines 329-338). In contrast, C1q is directly involved in tagging synapses to be pruned, while C4 may be involved in other schizophrenia related processes that occur in MIA, making these signals more widespread and specific to the model we were using. However, we could also speculate that the C1q and C4 signals could also have been stronger if we measured specific brain regions, and may be disrupted earlier.

“By measuring protein levels in whole brain, we may have washed out region specific signals. Previous groups found differences in spine dynamics relative to region and neuron type [20]. Complement expression changes in affected synapses isolated to specific regions may be proportionately masked when including large amounts of brain tissue from regions where no changes are occurring. This may also explain a lack of significant differences seen in C3. C3 activation is the convergence point of the classical, lechtin, and alternative complement pathways [21]. As C3 is participating in many functions other than synaptic pruning, alterations in C3 especially may not be detectable at the level of the entire brain, and may only be appear in certain regions or even synapse types. Analyzing specific brain regions, or using single cell sequencing, could be informative.”

In my opinion, the components from the statistical analysis placed in the supplementary materials will improve the reading of the description of the results.

We have moved the two panels from Supplementary Figure S1 into Figure 1 so they are now included in the main results.

Description to Figure 3. I would suggest changing the title to: MIA activation increased expression of the microglia phagocytosis marker CD68 in an age-dependent manner.

We have changed the title of Figure 3 to match this suggestion.

Reviewer 2 Report

Comments and Suggestions for Authors

This animal study examined the effects of pre-natal immune insult by polyinosinic:polycytidylic acid (polyI:C) on the neurodevelopmental processes, including microglial regulation of synaptic spines and complement signaling. The results of this study indicated that the disruptions in the pre-natal environment lead to alterations in multiple dynamic processes through to post-natal development.

This study examined an important issue and provided knowledge to the etiology of mental disorders.

The authors may consider revise their manuscript to improve its rigor.

1.      The current title is “Maternal Immune Activation Causes Deregulation of Microglia and the Complement System in Mice Leading to Decreased Synaptic Spine Density.” Many factors can cause maternal immune activation and result in various consequences. The authors may consider to specify the title by adding the method of activation used in this study into the title.

2.      Although this study found reduced dendritic spine density in the somatosensory cortex starting at 3-weeks-of-age with requisite increases in microglial synaptic pruning and phagocytosis and dysregulation in complement protein expression persisting into adulthood, the connections with behavioral changes similar with mental diseases were not explored. Further explanation is needed.

3.      The authors stated that disruptions in the schizophrenia-related phenotype, prepulse inhibition (PPI), vary in polyI:C MIA models depending on the gestational day of exposure. Further discussion on the results of this study based on the intervention procedure used in this study is needed.

4.      The reason for selecting male mice warrants explanation.

Author Response

We thank the reviewer for their helpful comments and suggestions. Our response and edits are highlighted below:

  1. The current title is “Maternal Immune Activation Causes Deregulation of Microglia and the Complement System in Mice Leading to Decreased Synaptic Spine Density.” Many factors can cause maternal immune activation and result in various consequences. The authors may consider to specify the title by adding the method of activation used in this study into the title.

We agree and have added the MIA method to the title of the manuscript. It now reads: “PolyI:C Maternal Immune Activation on E9.5 Causes Deregulation of Microglia and the Complement System in Mice Leading to Decreased Synaptic Spine Density”.

  1. Although this study found reduced dendritic spine density in the somatosensory cortex starting at 3-weeks-of-age with requisite increases in microglial synaptic pruning and phagocytosis and dysregulation in complement protein expression persisting into adulthood, the connections with behavioral changes similar with mental diseases were not explored. Further explanation is needed.

We collected some behavioural data from the mice used in this study. We chose to measure prepulse inhibition, as it is commonly measured in previous studies [1-6]. We found significant age × MIA interactions on global % PPI (69dB, 73dB, and 81dB prepulse intensities, 120dB pulse, 60dB background) (F3,34 = 3.69, p = 0.021*, η2p = 0.09 [0; 1]) and in baseline startle amplitude (F3,33 = 3.58, p = 0.024*, η2p = 0.25 [0.02; 1]). All of the age groups had normal PPI except the adult 12 wo mice which had significantly lower PPI (t(34) = 3.15, p = 0.0034**, d = 2.23 [0.69; 3.77]). Unfortunately, CD-1 mice are known to have congenital deafness, and we had reason to believe this was affecting the adult age group as they had lower average startle response than expected. 8 mice didn’t respond to the sound stimulus at all and were likely deaf (1 saline 8 wo, 3 polyI:C 8 wo, 2 saline 12 wo, 2 polyI:C 12 wo; excluded from previous statistics). Mice in the older two groups were significantly more likely to be unresponsive (X2 (3, N = 50) = 8.9, p = 0.031*), but 8 and 12 wo mice did not significantly differ by MIA exposure in stimulus responsiveness (X2 (1, N = 26) = 0.18, p = 0.67). For this reason, we felt the integrity of this result was questionable and chose to cut the behavioural data from the final manuscript.

  1. Woods, R. M.; Lorusso, J. M.; Fletcher, J.; ElTaher, H.; McEwan, F.; Harris, I.; Kowash, H. M.; D'Souza, S. W.; Harte, M.; Hager, R.; Glazier, J. D., Maternal immune activation and role of placenta in the prenatal progra mming of neurodevelop-mental disorders. Neuronal Signal 2023, 7, (2), NS20220064.
  2. Ding S, Hu Y, Luo B, Cai Y, Hao K, Yang Y, et al. Age-related changes in neuroinflammation and prepulse inhibition in offspring of rats treated with Poly I:C in early gestation. Behav Brain Funct. 2019;15(1):3.
  3. Ozawa K, Hashimoto K, Kishimoto T, Shimizu E, Ishikura H, Iyo M. Immune activation during pregnancy in mice leads to dopaminergic hyper function and cognitive impairment in the offspring: a neurodevelopment al animal model of schizophrenia. Biol Psychiatry. 2006;59(6):546-54.
  4. Wolff AR, Bilkey DK. The maternal immune activation (MIA) model of schizophrenia produces p re-pulse inhibition (PPI) deficits in both juvenile and adult rats but these effects are not associated with maternal weight loss. Behav Brain Res. 2010;213(2):323-7.
  5. Howland JG, Cazakoff BN, Zhang Y. Altered object-in-place recognition memory, prepulse inhibition, and locomotor activity in the offspring of rats exposed to a viral mimetic during pregnancy. Neurosci. 2012;201:184-98.
  6. Zhao X, Erickson M, Mohammed R, Kentner AC. Maternal immune activation accelerates puberty initiation and alters m echanical allodynia in male and female C57BL6/J mice. Dev Psychobiol. 2022;64(5):e22278.

3.      The authors stated that disruptions in the schizophrenia-related phenotype, prepulse inhibition (PPI), vary in polyI:C MIA models depending on the gestational day of exposure. Further discussion on the results of this study based on the intervention procedure used in this study is needed.

We have expanded on the explanation of this paper in the introduction (lines 53-62). It now reads: “For example, early (mouse gestational day E9) but not late (E17) polyI:C exposure causes deficits in the schizophrenia-associated reflex behaviour known as prepulse inhibition (PPI) and caused reductions in prefrontal dopamine D1 receptor expression; whereas, late but not early exposure disrupted working memory and hippocampal NMDA-type glutamate receptors [10]. There are also exposure-timing differences on neurodevelopmental effects of MIA, with late exposure leading to marked neuroanatomical changes and oxidative stress in E18 embryos but fewer detectable differences in adolescent and adult offspring; whereas, early exposure (E9) led to increased E18 apoptotic cells with significant neuroanatomical differences in adolescent offspring [11, 12].”

Added references:

  1. Guma, E.; Bordignon, P. d. C.; Devenyi, G. A.; Gallino, D.; Anastassiadis, C.; Cvetkovska, V.; Barry, A. D.; Snook, E.; Ger-mann, J.; Greenwood, C. M. T.; Misic, B.; Bagot, R. C.; Chakravarty, M. M., Early or Late Gestational Exposure to Maternal Immune Activation Alters Neurodevelopmental Trajectories in Mice: An Integrated Neuroimaging, Behavioral, and Tran-scriptional Study. Biol Psychiatry 2021, 90, (5), 328-341.
  2. Guma, E.; Bordeleau, M.; González Ibáñez, F.; Picard, K.; Snook, E.; Desrosiers-Grégoire, G.; Spring, S.; Lerch, J. P.; Nieman, B. J.; Devenyi, G. A.; Tremblay, M.-E.; Chakravarty, M. M., Differential effects of early or late exposure to prenatal maternal immune activation on mouse embryonic neurodevelopment. PNAS 2022, 119, (12), e2114545119-e2114545119.

4.      The reason for selecting male mice warrants explanation.

As this was a time series experiment, we had a large number of groups already (2 treatments X 5 ages). Adding sex as a factor would double the number of groups. To compound this, previous studies have reported sex interactions when measuring microglia in MIA [e.g., ref 39]. Our study would be far too underpowered to detect the expected sex interactions across 20 groups. We chose to analyze males so our results would be comparable to previous publications that only used males [33-35, 37, 40-41].

  1. Hui, C. W.; Vecchiarelli, H. A.; Gervais, É.; Luo, X.; Michaud, F.; Scheefhals, L.; Bisht, K.; Sharma, K.; Topolnik, L.; Tremblay, M.-È., Sex Differences of Microglia and Synapses in the Hippocampal Dentate Gyrus
  2. Abazyan, B.; Nomura, J.; Kannan, G.; Ishizuka, K.; Tamashiro, K. L.; Nucifora, F.; Pogorelov, V.; Ladenheim, B.; Yang, C.; Krasnova, I. N.; Cadet, J. L.; Pardo, C.; Mori, S.; Kamiya, A.; Vogel, M. W.; Sawa, A.; Ross, C. A.; Pletnikov, M. V., Prenatal interaction of mutant DISC1 and immune activation produces ad ult psychopathology. Biol Psychiatry 2010, 68, (12), 1172-81. of Adult Mouse Offspring Exposed to Maternal Immune Activation. Front Cell Neurosci 2020, 14.
  3. Baharnoori, M.; Brake, W. G.; Srivastava, L. K., Prenatal immune challenge induces developmental changes in the mor-phology of pyramidal neurons of the prefrontal cortex and hippocampus in rats. Schizophr Res 2009, 107, (1), 99-109.
  4. Lin, Y.-L.; Wang, S., Prenatal lipopolysaccharide exposure increases depression-like behaviors and reduces hippocampal neurogenesis in adult rats. Behav Brain Res 2014, 259, 24-34.
  5. Soumiya, H.; Fukumitsu, H.; Furukawa, S., Prenatal immune challenge compromises development of upper-layer but not deeper-layer neurons of the mouse cerebral cortex. J Neurosci Res 2011, 89, (9), 1342-50.
  6. Mattei, D.; Ivanov, A.; Ferrai, C.; Jordan, P.; Guneykaya, D.; Buonfiglioli, A.; Schaafsma, W.; Przanowski, P.; Deu-ther-Conrad, W.; Brust, P.; Hesse, S.; Patt, M.; Sabri, O.; Ross, T. L.; Eggen, B. J. L.; Boddeke, E. W. G. M.; Kaminska, B.; Beule, D.; Pombo, A.; Kettenmann, H.; Wolf, S. A., Maternal immune activation results in complex microglial transcriptome sig-nature in the adult offspring that is reversed by minocycline treatment. Transl Psychiatry 2017, 7, (5), e1120.
  7. Han, M.; Zhang, J.-C.; Hashimoto, K., Increased Levels of C1q in the Prefrontal Cortex of Adult Offspring af ter Maternal Immune Activation: Prevention by 7,8-Dihydroxyflavone. Clin Psychopharmacol Neurosci 2017, 15, (1), 64-67.